# Stability and applicability of the leaf value model for variable nitrogen application based on SPAD value in rice

Jie Li[1,2], Yuehua Feng[1,3]*, Xiaoke Wang[1], Jinfeng Peng[1], Dinghua Yang[2], Guiling Xu[1], Qiangxin Luo[1], Lingli Wang[1], Da Ou[1], Wei Su[1]

**1** College of Agronomy, Guizhou University, Guiyang, China, **2** Qiandongnan Vocational and Technical College for Nationalities, Kaili, China, **3** Laboratory of Plant Resource Conservation and Germplasm Innovation in Mountainous Region (Ministry of Education), Guizhou University, Guiyang, China

* fengyuehua2006@126.com

## Abstract

Many fertilization models have been created to scientifically determine the amount of fertilization. With the same purpose, we constructed a nitrogen (N) application model, the leaf value model, which can make N fertilizer decisions in a timely, fast and nondestructive manner during rice planting. However, only one area ($A_1$, Jiuzhou Town, Xixiu District, Guizhou Province) and one cultivar (Qyou6) were involved in the construction of the leaf value model. Its stability and applicability could not be well evaluated. Thus, we chose another area ($A_2$, Jiuzhou Town, Huangping County, Guizhou Province) in Guizhou Province and carried out the experiment by using four cultivars (Nie5you5399, Qyou6, Yixiangyou2115 and Zhongzheyou8) for the leaf value model construction. Compared with the average value of apparent total N uptake ($Nz$) obtained in 2 years in the A1 area, that in the Qyou6 leaf value model in the A2 area increased by 12%, reaching 635.72 kg ha$^{-1}$, whereas the corresponding target yield changed slightly, reaching 10,999.90 kg ha$^{-1}$. Simultaneously, the linear relationship between several good SPAD value-derived indexes ($Ys$) and apparent N supply of the field ($Nx$) was still significant or extremely significant in the Qyou6 leaf value model. Compared with the $A_1$ area, it slightly differed, and the $R^2$ of $SPAD_{L1}$ was higher than that of $SPAD_{L3 \times L4/mean}$. In the leaf value model of the other three cultivars, the relationship between yield and $Nx$ and that between $Ys$ and $Nx$ were significant or extremely significant. The $Nz$ of Yixiangyou2115 and Zhongzheyou8 (618.33 and 617.76 kg ha$^{-1}$) were close to that of Qyou6 and the corresponding target yields were 10313.36 and 10301.99 kg ha$^{-1}$, respectively. The $Nz$ and target yield of Nie5you5399 were lowest at 546.63 and 10680.24 kg ha$^{-1}$, respectively. In general, this study showed that relationships used in the construction of leaf value model had certain stability and applicability to difference areas and cultivars. The leaf value model can be considered in N fertilizer decision-making of rice planting management.

**Data Availability Statement:** All relevant data are within the manuscript.

**Funding:** This work was supported by the National Natural Science Foundation of China (31360311, 31160263), the Subproject of Special fund project

for Scientific Research of Public Welfare Industry (Agriculture) (201503118-03), the Scientific and Technological Innovative Talents Team for Cultivation and Eco-Physiology Research of Featured Grain and Oil Crops in Guizhou Province (Grant no. Qiankehe Platform Talents [2019] 5613), the Talents Program of High Level and Innovative in Guizhou Province (Grant no. Qiankehe Platform Talents [2018]5632), the Construction Program of Biology First-class Discipline in Guizhou Province (GNYL [2017]009) and the Agricultural Scientific and Technological Project in Guizhou Province (Qiankehe NY[2013] 3005).

**Competing interests:** The authors have declared that no competing interests exist.

## Introduction

Nitrogen (N) application is one of the most important artificial measures in crop planting management and is closely related to yield and environment. If the N application rate is too low to meet the needs of rice plant growth, then the ideal yield cannot be obtained; too much N application rate will lead to waste of resources and environmental pollution [1–2]. For this reason, researchers have constructed dozens of fertilization models. The construction of fertilization models is based on the diagnosis of nutrition and demand for yield. Nutrition diagnoses include the determination of soil nutrients and the analysis of plant nutrition indexes [3]. Previous research focused on the analysis of soil nutrients, with development of instruments, and later period focused on the analysis of plant nutrition indexes. In plant nutrition diagnosis, especially N nutrition diagnosis, physical diagnosis methods gradually replaced the traditional chemical diagnosis methods, such as the application of hyperspectral equipment, SPAD meter, Greenseeker and digital camera [4–8]. Physical diagnosis methods, such as the SPAD meter application, could evaluate the N nutrition in a timely, quick and non-destructive manner [9]. The SPAD value represents the relative chlorophyll content, which can be related to photosynthesis and N nutrition status. Thus, it is widely used in plant research.

Many reports on N diagnosis based on the SPAD value are available. These reports focus on the determination site, determination period and relationship with N nutrition [10]. In determining the N application rate based on the SPAD value, the threshold model is used. If the actual measured SPAD value exceeds the set threshold SPAD value, fertilization or a small amount of fertilizer will be applied; otherwise, fertilization will not be applied [11–13]. The threshold model cannot well quantify the amount of N fertilizer, which is not easy to obtain. However, some fertilization models, such as the target yield model, involve some parameters, e.g. soil N supply and fertilizer recovery efficiency, which are variable and difficult to calibrate [14]. Therefore, in 2015–2016, based on a large number of experiments, we constructed an N application model, leaf value model, which had a linear relationship between the SPAD value and N application rate,and could guide N topdressing in rice [15]. The leaf value model can be used to estimate the N content of soil quickly and simply by the SPAD meter, and it also has the intuitiveness of the fertilizer effect function model.

At that time of first construction, however, only one area ($A_1$, Jiuzhou Town, Xixiu District, Guizhou Province) and one cultivar (Qyou6) were involved in the construction of the leaf value model. We could not determine whether the relationship involved was tenable in another area and other cultivars. We could not further evaluate the model's stability and applicability. Therefore, in this study, we chose another experiment site ($A_2$, Jiuzhou Town, Huangping County, Guizhou Province) and constructed the leaf value model for four cultivars to test whether the relationship in model are still significant or extremely significant. Meanwhile, we carried out the variable N application experiment based on the leaf value model at the former test site to reinvestigate the variable N application effect of the leaf value model in that place.

## Materials and methods

### Experiment site description

Field experiment was conducted in Jiuzhou town ($A_1$), Xixiu District, Guizhou Province, China (26°10′58.8″–26°17′56.4″N, 106°2′42″–106°11′42″E) and Jiuzhou town ($A_2$), Huangping County, Guizhou Province, China (26°54′54″–27°5′49.2″N, 107°38′45.6″–107°51′39.6″E). The climate of $A_1$ and $A_2$ in the field test area of this study belongs to the north subtropical monsoon humid climate. The altitude, annual average temperature, a frost-free period and

annual rainfall in $A_1$ and $A_2$ are as follows: 1275 m, 14.3°C, 269.9 days and 1360 mm; and 698 m, 15.7°C, 268 days and 1200 mm, respectively. In 2015–2016, we conducted experiments in $A_1$ and constructed the leaf value model for the first time. One paddy field (F1) was used in in A1, and five paddy fields (F2, F3, F4, F5 and F6) were used in A2. The environmental differences between the two areas were obvious, and both areas have large paddy fields in Guizhou Province.

## Experiment design

In F1 of A1, random block design was used with three treatments, blank control, average N fertilization and variable N fertilization. The blank control did not apply N fertilizer, and three plots were available. N fertilizer was split-applied at a rate of 150 kg N ha$^{-1}$ as 35% basal and 20% at early-tillering in variable N fertilization. When fertilization was performed at the panicle fertilizer stage, the leaf value model was used to determine the top-dressing fertilizer rate. Five plots existed in variable N fertilization. Different from variable N fertilization, the N application rate of panicle fertilizer in average N fertilization was the average rate of the N fertilizer of five plots in variable N fertilization. Four plots were observed in average N fertilization. All plots were 12 m$^2$ in F6.

In each paddy field of A2, the experiments were in a split-plot design with N application rates as the main plots and cultivars as the subplots. The N application rate was N0 (0 kg ha$^{-1}$), N1 (75 kg ha$^{-1}$), N2 (150 kg ha$^{-1}$) and N3 (225 kg ha$^{-1}$), and the rice cultivars included Nie5you5399, Qyou6, Yixiangyou2115 and Zhongzheyou8. The N fertilizer was split-applied in treatments with 35% as basal, 20% at early tillering (7 days after transplanting), 30% at flower-promoting fertilizer and 15% at flower fertilizer. The sizes of the main plot were 37.38, 34.20, 43.80, 33.60 and 34.20 m$^2$ in F2, F3, F4, F5 and F6, respectively. The experiment was replicated twice in F2 and thrice in other fields. However, in the course of the experiment, all field plots were only repeated twice due to the relatively large workload.

In addition, Phosphorus (P) and potassium (K) fertilizers were applied in the same manner in all treatments. P fertilizer was applied as basal at a rate of 96 kg P$_2$O$_5$ ha$^{-1}$, and K fertilizer was split-applied at a rate of 180 kg K$_2$O ha$^{-1}$ as 50% basal and 50% booting. Urea, superphosphate and potassium chloride were used as N, P and K fertilizers, respectively.

## Soil and crop management

Before ploughing, soil samples were taken from the upper 20 cm of the soil to determine the chemical properties of six paddy fields. The soil chemical properties are listed in Table 1. Seeds were sown on late April and transplanting were conducted on late May. Before transplanting, the basal fertilizer was applied according to the fertilization design. When transplanting, the rice seedlings in A1 and A2 were transplanted at a hill spacing of 16.7 cm × 30 cm and 20

**Table 1. Soil chemical properties of the 0–20 cm layer of the experimental plots.**

| Field number | Total N (g kg$^{-1}$) | Organic Matter (g kg$^{-1}$) | Alkali-hydrol N (mg kg$^{-1}$) | Olsen-P (mg kg$^{-1}$) | Avail-K (mg kg$^{-1}$) | pH |
|---|---|---|---|---|---|---|
| F1 | 2.75 | 35.34 | 176.89 | 8.35 | 235.28 | 6.05 |
| F2 | 2.61 | 20.60 | 189.38 | 3.43 | 87.55 | 4.82 |
| F3 | 2.59 | 27.71 | 194.03 | 13.43 | 99.15 | 5.28 |
| F4 | 1.63 | 17.87 | 126.93 | 2.81 | 71.03 | 5.22 |
| F5 | 2.54 | 25.88 | 165.09 | 5.97 | 161.70 | 5.24 |
| F6 | 1.54 | 14.70 | 107.43 | 13.90 | 112.94 | 6.38 |

cm × 30 cm, with one seedling per hill. 7 days after transplanting, tillering fertilizer was applied. And panicle fertilizer was applied according to the fertilization design. It was fine management of water and fertilizer during rice growth, and insects were intensively controlled by chemicals to avoid yield loss.

## Plant sampling

At maturity, according to the standard of the average tiller number (10 points), the plants were sampled from six hills in each subplot. The plant samples were separated into straw (including rachis) and filled and unfilled spikelets and oven-dried at 80°C to constant weight. The total biomass production was the summation of straw, filled and unfilled spikelet dry matter. Then, the N content of straw (including rachis), filled and unfilled spikelets was determined.

## Introduction of leaf value model

According to the previous studies (Li *et al.*, 2017), a general expression for leaf value model is

$$Nw = Nz - [(Ys - b) \div k - Ng] \tag{1}$$

Where:

$Nw$, total N application rate (kg ha$^{-1}$);

$Nz$, apparent total N uptake (kg ha$^{-1}$);

$Ys$, SPAD value derivative index;

$k$ and $b$, the slope and origin of the linear relationship between the apparent N supply of field ($Nx$) and $Ys$;

$Ng$, N application rate before topdressing (kg ha$^{-1}$).

$Nx$ is the sum of the $Nt$ (kg ha$^{-1}$) and $Ng$. $Nt$ is the product of alkali-hydrol N and soil density ($2.25\times10^6$ kg ha$^{-1}$). $Nz$ is calculated by the curve fitting relationship between the yield and $Nx$ and is the $Nx$ that reaches the target yield at heading. $Ys$ was obtained by processing the SPAD value of the rice canopy leaves through certain mathematical relations. In addition to the SPAD value of a single leaf, three $Yss$ were used in this study, that is, SPAD$_{average}$ (average SPAD value of the rice canopy leaves), SPAD$_{L2\times L3/average}$ (product of SPAD value of the second leaf and SPAD value of the third leaf divided by the average SPAD value of the rice canopy leaves), SPAD$_{L3\times L4/average}$ (product of the SPAD value of the third leaf and the SPAD value of the fourth leaf divided by the average SPAD value of the rice canopy leaves).

## Measurements and calculations

**SPAD measurements.** At jointing, five main stems of rice were randomly selected from each plot, and the SPAD value of the three fully expanded leaves at the top was determined by the SPAD meter (SPAD-502, Minolta Camera Co., Osaka, Japan), whereas the SPAD value of the four leaves at heading was determined. From the top down, the heart leaf of the main stem with more 80% extension was divided as the top 1 leaf (L$_1$). When the heart leaf did not reach 80% extension, the leaf as its next leaf is divided as L$_1$. During measurement, the narrow side on both sides of the main vein was selected for the measurement, and it was 1/2 of the width of narrow side of the leaf. Three measurement sites were available on the narrow side of a leaf, i.e. 1/2 of the length of the leaf and 3 cm up and down.

**Yield measurements.** In A1, The grain yield was determined from 90 hills in each plot and adjusted to the standard moisture content of 0.135 g H$_2$O g$^{-1}$ at maturity. In A2, as the yield data, the grain yield was determined from 40 hills in each subplot and adjusted to the standard moisture content of 0.135 g H$_2$O g$^{-1}$.

**N application rate of variable N fertilization and average N fertilization in A1.** At jointing, SPAD$_{L3}$ was measured as Y$s$. Then, the total N application rate of each plot was calculated through the leaf value model determined by the test data of 2015–2016:

$$N_w = 560.99 - [(Y_s - 24.00) \div 0.0346 - 82.50] \qquad (2)$$

The results showed that the final N application rate of the five plots in F1 was 153.12, 175.09, 227.11, 238.48 and 182.99 kg ha$^{-1}$. And the N application rate of each plot in average N fertilization was 195.36 kg N ha$^{-1}$.

**N use efficiency.** N agronomic efficiency (kg grain kg$^{-1}$ N) = (grain yield in N fertilization condition—grain yield in N-unfertilized condition)/N application rate, N recovery efficiency (%) = 100 × (total aboveground plant N accumulation in the plot received N fertilizer—total aboveground plant N accumulation in the zero-N control)/N application rate, N partial productivity (kg grain kg$^{-1}$ N) = grain yield in N fertilization condition/N application rate, N contribution rate (%) = 100 × (grain yield in N fertilization condition—grain yield in N-unfertilized condition)/grain yield in N fertilization condition.

**Harvest index.** 100 × filled spikelet weight/total biomass.

## Statistical analysis

Data were organized and analyzed by Microsoft Excel 2010. The SPAD value data in all plots was the average value of 15 data (3 sites on one leaf, 5 leaves in total). The coefficient of determination ($R^2$) was used to evaluate the relationship between yield and N$x$, and between Y$s$ and N$x$ by regression analysis (P = 0.01, P = 0.05). The maximum value, minimum value, average value, standard deviation (SD) and coefficient of variation (CV) were cited to compare and evaluate the effect of variable N fertilization and average N fertilization.

## Results

### Relationship between yield and Nx

In A$_2$ area, an extremely significant curve fitting relationship was observed between the yield of four rice cultivars and N$x$ (Fig 1). According to this relationship, the target yields of Nie5you5399, Qyou6, Yixiangyou2115 and Zhongzeyou8 were 10680.24, 10999.90, 10313.36 and 10301.99 kg ha$^{-1}$, respectively. The corresponding values of N$x$, that was their N$z$ values, were 546.63, 635.72, 618.33 and 617.76 kg ha$^{-1}$, respectively. The target yield and N$z$ of Yixiangyou2115 were close to those of Zhongzeyou8. N$z$ of Nie5you5399 was 14.01%, 11.60% and 11.51% lower than that of other cultivars. Whereas its target yield was only lower than those of Qyou6 about 2.91%, and was 3.56% and 3.67% higher than those of Yixiangyou2115 and Zhongzeyou8, respectively. In general, the target yields of the four cultivars were almost the same.

### Relationship between Ys and Nx

Similarly, in A$_2$ area, whether at jointing or heading, significant or extremely significant relationships were observed between several Y$s$s of the four rice cultivars and N$x$ (Table 2). Yixiangyou8 all reached an extremely significant relationship at jointing and heading. In addition, $R^2$ of SPAD$_{L1}$ and SPAD$_{L2}$ was higher at jointing than other Y$s$s, whereas it was SPAD$_{L3}$ and SPAD$_{average}$ at heading. As can be seen from the slope and origin of the linear fitting equation, slope at heading was generally higher than at jointing, and the origin was smaller at heading than at jointing. On the whole, $R^2$ at heading was higher at heading than at jointing. These results were consistent with the experiment results of A$_1$ in 2015 and 2016.

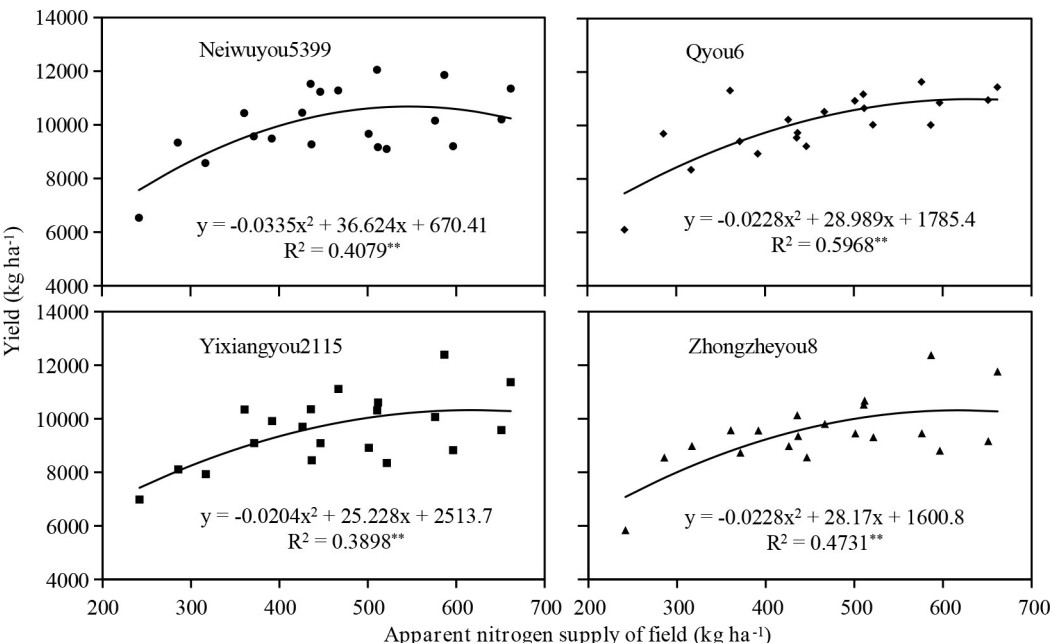

**Fig 1. Curve fitting between the apparent N supply (N$x$) and yield at heading.** ** indicates extremely significant correlation at 0.01 probability level. N$x$ is the sum of the N$t$ (kg ha−1) and N$g$. N$t$ is the product of alkali-hydrol N and soil density ($2.25×10^6$ kg ha$^{-1}$).

**Table 2. Linear fitting relationship between the apparent N supply (N$x$) and SPAD value derivative index (Y$s$) at the jointing and heading in rice.**

| Growth stage | Derivative index | Fitting equation | | | |
|---|---|---|---|---|---|
| | | Nie5you5399 | Qyou6 | Yixiangyou2115 | Zhongzheyou8 |
| Jointing stage | SPAD$_{L1}$ | Y$s$ = 0.0100N$x$+39.066 $R^2$ = 0.2629* | Y$s$ = 0.0114 N$x$ +38**.266 $R^2$ = 0.3601** | Y$s$ = 0.0137 N$x$ +34.570 $R^2$ = 0.3364** | Y$s$ = 0.0142 N$x$ +36.596 $R^2$ = 0.3471** |
| | SPAD$_{L2}$ | Y$s$ = 0.0111 N$x$ +39.238 $R^2$ = 0.2107* | Y$s$ = 0.0113 N$x$ +38.367 $R^2$ = 0.2168* | Y$s$ = 0.0168 N$x$ +34.633 $R^2$ = 0.3477** | Y$s$ = 0.0183 N$x$ +36.483 $R^2$ = 0.4218** |
| | SPAD$_{L3}$ | Y$s$ = 0.0139 N$x$ +38.605 $R^2$ = 0.2271* | Y$s$ = 0.0148 N$x$ +36.951 $R^2$ = 0.2938* | Y$s$ = 0.0165 N$x$ +35.872 $R^2$ = 0.2318* | Y$s$ = 0.0205 N$x$ +35.768 $R^2$ = 0.3740** |
| | SPAD$_{average}$ | Y$s$ = 0.0117 N$x$ +38.970 $R^2$ = 0.2372* | Y$s$ = 0.0125 N$x$ +37.861 $R^2$ = 0.2993* | Y$s$ = 0.0157 N$x$ +35.025 $R^2$ = 0.3064* | Y$s$ = 0.0177 N$x$ +36.282 $R^2$ = 0.3902** |
| | SPAD$_{L2×L3/average}$ | Y$s$ = 0.0133 N$x$ +38.871 $R^2$ = 0.2062* | Y$s$ = 0.0136 N$x$ +37.451 $R^2$ = 0.2286* | Y$s$ = 0.0177 N$x$ +35.460 $R^2$ = 0.2667* | Y$s$ = 0.0212 N$x$ +35.957 $R^2$ = 0.4014** |
| Heading stage | SPAD$_{L1}$ | Y$s$ = 0.0183 N$x$ +35.828 $R^2$ = 0.2788* | Y$s$ = 0.0091 N$x$ +39.577 $R^2$ = 0.2058* | Y$s$ = 0.0239 N$x$ +26.688 $R^2$ = 0.5007** | Y$s$ = 0.0170 N$x$ +30.987 $R^2$ = 0.6960** |
| | SPAD$_{L2}$ | Y$s$ = 0.0116 N$x$ +37.328 $R^2$ = 0.4203** | Y$s$ = 0.0356 N$x$ +26.927 $R^2$ = 0.5134** | Y$s$ = 0.0190 N$x$ +28.481 $R^2$ = 0.5830** | Y$s$ = 0.0210 N$x$ +28.767 $R^2$ = 0.7348** |
| | SPAD$_{L3}$ | Y$s$ = 0.0288 N$x$ +29.511 $R^2$ = 0.5032** | Y$s$ = 0.0205 N$x$ +31.238 $R^2$ = 0.6571** | Y$s$ = 0.0247 N$x$ +27.097 $R^2$ = 0.6164** | Y$s$ = 0.0321 N$x$ +25.048 $R^2$ = 0.6062** |
| | SPAD$_{L4}$ | Y$s$ = 0.0265 N$x$ +28.301 $R^2$ = 0.4437** | Y$s$ = 0.0303 N$x$ +24.050 $R^2$ = 0.4756** | Y$s$ = 0.0417 N$x$ +19.359 $R^2$ = 0.5900** | Y$s$ = 0.0324 N$x$ +24.070 $R^2$ = 0.7026** |
| | SPAD$_{average}$ | Y$s$ = 0.0213 N$x$ +32.742 $R^2$ = 0.5508** | Y$s$ = 0.0239 N$x$ +30.448 $R^2$ = 0.6346** | Y$s$ = 0.0273 N$x$ +25.406 $R^2$ = 0.6831** | Y$s$ = 0.0256 N$x$ +27.218 $R^2$ = 0.7611** |
| | SPAD$_{L3×L4/average}$ | Y$s$ = 0.0332 N$x$ +25.418 $R^2$ = 0.4926** | Y$s$ = 0.0274 N$x$ +24.713 $R^2$ = 0.4147** | Y$s$ = 0.0390 N$x$ +21.063 $R^2$ = 0.5648** | Y$s$ = 0.0385 N$x$ +22.031 $R^2$ = 0.7203** |

* indicates significant correlation at 0.05 probability level.

** indicates extremely significant correlation at 0.01 probability level.

**Table 3. General mathematical expression of the leaf value model at the jointing and heading in the four rice cultivars.**

| Cultivar | Jointing stage | Heading stage |
|---|---|---|
| Nie5you5399 | N$w$ = 3323.97–71.9424Y$s$+N$g$ | N$w$ = 1571.32–34.7222 Y$s$+N$g$ |
| Qyou6 | N$w$ = 3132.41–67.5676 Y$s$+N$g$ | N$w$ = 2159.52–48.7805 Y$s$+N$g$ |
| Yixiangyou2115 | N$w$ = 2792.39–60.6061 Y$s$+N$g$ | N$w$ = 1715.37–40.4858 Y$s$+N$g$ |
| Zhongzheyou8 | N$w$ = 2362.54–48.7805 Y$s$+N$g$ | N$w$ = 1398.07–31.1526 Y$s$+N$g$ |

Nw, total N application rate (kg ha$^{-1}$); Ys, SPAD value derivative index; Ng, N application rate before topdressing (kg ha$^{-1}$).

## Mathematical expression of the leaf value model for the four rice cultivars

According to the relationship between N$x$ and yield, N$x$ and SPAD$_{L3}$, the leaf value models of the four rice cultivars were constructed (Table 3). When SPAD$_{L3}$ increased by 1, the top-dressing rates of Nie5you5399, Qyou6, Yixiangyou2115 and Zhongzeyou8 would decrease by 71.94, 67.57, 60.61 and 48.78 kg ha$^{-1}$ at jointing and by 34.72, 48.78, 40.49 and 31.15 kg ha$^{-1}$ at heading, respectively. Moreover, when SPAD$_{L3}$ was 46.20, 46.36, 46.07 and 48.43, respectively, N fertilizer could not be applied at jointing in the four rice cultivars.

## Effect of variable N application based on the leaf value model

As illustrated in Table 4, The yield of variable N fertilization was 4.39% higher than that of average N fertilization, and the CV of yield of variable N fertilization was significantly lower than that of average N fertilization. In harvest index, variable N fertilization was slightly higher than average N fertilization. Among N use efficiency, the N agronomic efficiency, N partial productivity and N contribution rate of variable N fertilization were 26.98%, 7.52% and 21.05% higher than that of average N fertilization, respectively. In general, when the leaf value model was used to guide the implementation of variable N fertilization, the differences in yield, harvest index, N agronomic efficiency, N recovery efficiency and N contribution rate

**Table 4. Yield, harvest index and N use efficiency under the leaf value model based on the variable rate fertilization in Qyou6.**

| Item | Treatment | Maximum | Minimum | Average | SD | CV |
|---|---|---|---|---|---|---|
| Yield (kg ha$^{-1}$) | Average-rate | 9703.72 | 8041.08 | 9042.08 | 801.31 | 0.09 |
| | Variable-rate | 9923.67 | 8901.83 | 9439.18 | 428.75 | 0.05 |
| Harvest index (%) | Average-rate | 57.97 | 53.04 | 55.79 | 2.12 | 0.04 |
| | Variable-rate | 59.02 | 54.84 | 56.77 | 1.71 | 0.03 |
| N agronomic efficiency (kg kg$^{-1}$) | Average-rate | 12.47 | 3.96 | 9.08 | 4.10 | 0.45 |
| | Variable-rate | 16.71 | 6.85 | 11.53 | 3.50 | 0.30 |
| N recovery efficiency (%) | Average-rate | 48.29 | 20.22 | 35.29 | 0.12 | 0.33 |
| | Variable-rate | 32.33 | 14.93 | 24.15 | 0.08 | 0.31 |
| N partial productivity (kg kg$^{-1}$) | Average-rate | 49.67 | 41.16 | 46.28 | 4.10 | 0.09 |
| | Variable-rate | 64.18 | 37.33 | 49.76 | 10.11 | 0.20 |
| N contribution rate (%) | Average-rate | 0.25 | 0.10 | 0.19 | 0.07 | 0.39 |
| | Variable-rate | 0.27 | 0.18 | 0.23 | 0.04 | 0.15 |

The grain yield as yield was adjusted to the standard moisture content of 0.135 g H$_2$O g$^{-1}$. Harvest index = 100 × filled spikelet weight/total biomass. N agronomic efficiency (kg grain kg$^{-1}$ N) = (grain yield in N fertilization condition—grain yield in N-unfertilized condition)/N application rate, N recovery efficiency (%) = 100 × (total aboveground plant N accumulation in the plot received N fertilizer—total aboveground plant N accumulation in the zero-N control)/N application rate, N partial productivity (kg grain kg$^{-1}$ N) = grain yield in N fertilization condition/N application rate, N contribution rate (%) = 100 × (grain yield in N fertilization condition—grain yield in N-unfertilized condition)/grain yield in N fertilization condition.

between area units could be reduced. Meanwhile, which could improve the yield, N agronomic efficiency, N partial productivity and N contribution rate.

## Discussions

### Determination of Nz

The relationship between the N$x$ and yield is an important basis in the construction of the leaf value model. Combined with the previous research, we can know that the relationship between the N$x$ and yield is extremely significant in the leaf value model of different cultivars in different areas and years.

In addition, we paid considerable attention to the change of N$z$ of different cultivars in different areas and years. Previous studies in A$_1$ area showed that N$z$ of Qyou6 in 2015 and 2016 was 575.27 and 546.71 kg ha$^{-1}$, respectively, with a difference of only 28.56 kg ha$^{-1}$. The corresponding target yield was 9264.93 and 11167.97 kg ha$^{-1}$, with a difference of 1903.04 kg ha$^{-1}$ [15]. In this study, N$z$ and yield of Qyou6 were 635.72 and 10999.90 kg ha$^{-1}$, respectively, which had approximately 13.32% and 7.67% higher than the average of 2 years in the A$_1$ area. Yield is affected by cultivar and environment, and soil N and artificial N supply are only one of the factors. Therefore, in previous studies, we tend to assume that N$z$ for cultivars to obtain its target yield was fixed, whereas the yield would be different due to different environmental factors, such as light and temperature. Previous experiment results also supported this hypothesis. To date, the results in A$_1$ and A$_2$ areas showed a 13.32% change in N$z$, whereas this did not affect our hypothesis. To make the application of leaf value model more convenient, we still think that this hypothesis is tenable and that N$z$ can be determined in 500–550 kg ha$^{-1}$. According to the 3-year experiment data of Qyou6 and experiment data of the three other cultivars, the average value of N$z$ was 590.07 kg ha$^{-1}$, and the corresponding yield was 10,454.73 kg ha$^{-1}$. When N$z$ was determined to be 550 and 500 kg ha$^{-1}$, the corresponding yield was 10390.40 and 10233.67 kg ha$^{-1}$. N$z$ decreased by 6.79% and 15.26%, but its corresponding yield only decreased by 0.62% and 2.11%, with minimal change.

### Selection of Ys

This study also showed that the relationship between N$x$ and Y$s$ also had stability and applicability. In the four rice cultivars, significant or extremely significant relationship between N$x$ and many Y$s$s at jointing and heading was observed, and the correlation at heading was still higher than at jointing. In combination with previous studies [15], under the same Y$s$, the slope and origin of the linear relationship between N$x$ and Y$s$ had certain changes in different areas, years and cultivars, but these changes were not very large. Moreover, in practical application, $k$ and $b$ could be obtained and corrected by only two treatments (with known N application rate). Here, we would like to highlight the selection of Y$s$. In previous studies [15], the L$_1$ we selected was semi-expanded at jointing. In this study, we changed this approach. As studied by Li [16], the heart leaf of the main stem with more than 80% extension was divided as L$_1$, or when the heart leaf does not reach 80% extension, the next leaf was divided as L$_1$. In addition, in this study, SPAD$_{L1}$ had higher R$^2$ than other Y$s$s, which was different from our previous studies. Some studies showed that the lower leaves were more effective than the upper leaves in diagnosis of rice plant N nutrition or in response to soil N supply [16–19]. In studies on rice [20], wheat [21,22], corn [23], cotton [24], barley [25] and potato [26], the top 1 leaf was also used for the measurement of the SPAD value. Here, as our previous research point of view, when the leaf value model was applied, Y$s$ was selected depending on the different purposes of users, such as convenience of measurement, size of correlation and N application rate [15]. Now, we prefer to use SPAD$_{L3}$.

## Appraisals of variable N application based on the leaf value model

According to the spatial variation of soil nutrients, precision agriculture divides the planting area into several small plots and determines the fertilization rate according to the soil N of each plot, thereby maximising the utilisation of the cultivated land and fertilizer resources [27]. Variable rate fertilization technology is the core of precision agriculture, and to a certain extent, the fertilization model determines the application effect of variable rate fertilization technology. The effect of variable N application based on the leaf value model showed that this model can evaluate N$t$ by measuring the SPAD value to obtain Y$s$ and can determine the N application rate according to the soil nutrient situation of the operating unit. Increasing the yield could also reduce the yield difference and increase the N agronomic efficiency, N partial productivity and N contribution rate.

The target yield fertilization model [28], the fertilizer effect function model [29] and the N nutrition index [30,31] are commonly used models that guide field crop N application. Compared with these models, the use of the leaf value model in the determination of N$z$ was similar to that of the fertilizer effect function model. The leaf value model also considers the soil nutrient status and other factors similar to those considered by the target yield model. Different from the diagnosis of plant N nutrition, the leaf value model is used to estimate the N$x$ by Y$s$. Our study showed that the leaf value model had stability and applicability.

In the construction and design of the leaf value model, the leaf value model at jointing was used for N topdressing in the season, and the leaf value model at heading was used to determine the base fertilizer and tiller fertilizer in the next season. For the application of the leaf value model at heading, further research is needed to show its application effect.

## Conclusions

The leaf value model, a linear relationship N fertilization model based on SPAD value constructed in 2015–2016, is a simple, quick and dynamic N decision model. For the parameters of the leaf value model and the relationship constructed, the results of this study show that there is extremely significant curve fitting relationship between N$x$ and yield, and significant or extremely significant linear fitting relationship is observed between N$x$ and Y$s$. Combined with the data of three years (2015, 2016, 2019), leaf value model has certain stability and applicability in different areas, years and cultivars. This N application model is worth considering and recommending in rice planting N decision-making. In addition, considering the change of Y$s$ effect, N$z$ and leaf value model coefficient, extensive research should be carried out under different environmental conditions.

## Author Contributions

**Conceptualization:** Yuehua Feng.

**Data curation:** Jie Li.

**Formal analysis:** Jie Li.

**Funding acquisition:** Yuehua Feng.

**Investigation:** Jie Li, Xiaoke Wang, Jinfeng Peng, Dinghua Yang, Guiling Xu, Lingli Wang, Da Ou, Wei Su.

**Methodology:** Jie Li, Yuehua Feng.

**Project administration:** Jie Li, Yuehua Feng, Xiaoke Wang, Qiangxin Luo.

**Resources:** Yuehua Feng.

**Supervision:** Yuehua Feng.

**Visualization:** Jie Li, Yuehua Feng.

**Writing – original draft:** Jie Li.

**Writing – review & editing:** Yuehua Feng.

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
