## [Decision Letter · Decision Letter 0]

24 Feb 2020

PONE-D-20-02735

Stability and applicability of the leaf value model for variable rate fertilization based on SPAD value in rice

PLOS ONE

Dear feng,

Thank you for submitting your manuscript to PLOS ONE. After careful consideration, we feel that it has merit but does not fully meet PLOS ONE’s publication criteria as it currently stands. Therefore, we invite you to submit a revised version of the manuscript that addresses the points raised during the review process.

We would appreciate receiving your revised manuscript by Apr 09 2020 11:59PM. To enhance the reproducibility of your results, we recommend that if applicable you deposit your laboratory protocols in protocols.io, where a protocol can be assigned its own identifier (DOI) such that it can be cited independently in the future. For instructions see: http://journals.plos.org/plosone/s/submission-guidelines#loc-laboratory-protocols

We look forward to receiving your revised manuscript.

Kind regards,

Bing Xue, Ph.D.

Academic Editor

PLOS ONE

Journal Requirements:

1. In your Methods section, please provide additional information regarding the permits you obtained for the work. Please ensure you have included the full name of the authority that approved the field site access and, if no permits were required, a brief statement explaining why.

3. We note you have included a table to which you do not refer in the text of your manuscript. Please ensure that you refer to Tables 2 & 4 in your text; if accepted, production will need this reference to link the reader to the Table.

Reviewers' comments:

Reviewer's Responses to Questions

**Comments to the Author**

1. Is the manuscript technically sound, and do the data support the conclusions?

Reviewer #1: Yes

Reviewer #2: Partly

2. Has the statistical analysis been performed appropriately and rigorously? 

Reviewer #1: Yes

Reviewer #2: No

3. Have the authors made all data underlying the findings in their manuscript fully available?

Reviewer #1: Yes

Reviewer #2: No

4. Is the manuscript presented in an intelligible fashion and written in standard English?

Reviewer #1: Yes

Reviewer #2: Yes

5. Review Comments to the Author

Reviewer #1: In this paper, a leaf value model was built based on SPAD value for nitrogen management in rice, and the method was innovative. However, considering the difference of soil environment in different rice varieties and planting areas, it was suggested to conduct further research.

Reviewer #2: It is interesting to develop the leaf value model to determine optimal N application rate. However, there are several main questions.

(1) The comparison of A2 with A1 is not suitable, A2 involves 4 cultivars and several field locations, A1 involves only one, I suggest to combine A1 and A2 together.

(2) How did the authors calculate the apparent N supply of filed in Fig. 1? Please explain in Fig. 1.

(3) In Fig.1, I did not see the relationship between actual N application rates and grain yields of rice at harvest, suggest to add.

(4) What is the relationship between the model you established and the adoption, I am confused. (5) Please add some more parameters in relation to rice yield components and their responses to N and SPAD values.

(6) If possible, please add some experimental photos of SPAD values with changing N application rates.

Together, I suggest major revisions. It cannot be accepted according to the present content.

Several minor revisions:

(1) The language needs to be improved further.

(2) The description of results “effect of variable N application based on the leaf value model” is too easy.

(3) In Table 3, please explain what are Nw, Ys and Ng.

(4) In Table 4, please explain how to calculate Yield, Harvest index, N agronomic efficiency, N recovery efficiency, N partial productivity and N contribution.

(5) Conclusions are too easy.

(6) Statistical Analysis is too easy, please add some more description.

(7) Please separate “Experiment Design” into several parts including such as experimental design, soil and crop management, plant sampling, calculation, to make the materials and methods more clear.

6. PLOS authors have the option to publish the peer review history of their article (what does this mean?). If published, this will include your full peer review and any attached files.

Reviewer #1: No

Reviewer #2: No

---

## [Author Response · Author response to Decision Letter 0]

22 Apr 2020

PONE-D-20-02735

Stability and applicability of the leaf value model for variable rate fertilization based on SPAD value in rice

PLOS ONE

Dear Xue,

Thank you very much for your work in the manuscript submission process. At the same time, thank two reviewers very much for the recognition and valuable suggestions. Their suggestions make our manuscript more excellent. We have submitted revised version of the manuscript in PLOS ONE and ensured that:

1.'Response to Reviewers', 'Revised Manuscript with Track Changes' and 'Manuscript' all were submitted.

2. When the manuscript is revised, we refer to PLOS ONE's style requirements.

3. In the methods section, we do not need to provide information about work permit. The research we have carried out does not involve ethics, safety and other factors. It is a fully liberalized research work.

4. We have put the title page into the main document.

5. Table 2 and 4 have been mentioned in the manuscript.

For this revision manuscript, we make sure that it is more perfect and meets the requirements of PLOS ONE, but we are still worried about the places that are not considered.

We are looking forward to your reply again. 

Yuehua Feng

Professor

College of Agronomy, Guizhou University, Guiyang, Guizhou, 550025, China.

Laboratory of Plant Resource Conservation and Germplasm Innovation in

Mountainous Region (Ministry of Education), Guizhou University, Guiyang 550025,

China.

Phone: +86-13984385198

Email: fengyuehua2006@126.com

Response to Reviewers

Response to Reviewers #1

Reviewer #1: In this paper, a leaf value model was built based on SPAD value for nitrogen management in rice, and the method was innovative. However, considering the difference of soil environment in different rice varieties and planting areas, it was suggested to conduct further research.

Response: This is indeed the case. Considering the complexity of the factors, the improvement and application of the leaf value model will be a topics of our main research. For this point, we also add some instructions in the "Conclusion" section of the current manuscript.

Response to Reviewers #2

Reviewer #2: The comparison of A2 with A1 is not suitable, A2 involves 4 cultivars and several field locations, A1 involves only one, I suggest to combine A1 and A2 together.

Response: The experiments of A1 and A2 were arranged to verify the stability and applicability of the model, but our statement in this paper is easy to cause the comparison of A2 with A1. For this reason, we accepted the opinions of the reviewer #2, readjusted the materials and methods with synthesizing them into one experiment. 

Reviewer #2: How did the authors calculate the apparent N supply of filed in Fig. 1? Please explain in Fig. 1.

Response: There is an explanation in Fig. 1.

Reviewer #2: In Fig.1, I did not see the relationship between actual N application rates and grain yields of rice at harvest, suggest to add.

Response: In Fig. 1, it is mainly to clarify the relationship between yield and apparent N supply of field (actual N application rate + apparent N supply of soil). Whether this relationship is established or not is crucial to the construction of leaf value model. 

Reviewer #2: What is the relationship between the model you established and the adoption, I am confused.

Response: The leaf value model linked the four aspects of soil condition, rice growth, N application rate and yield. But this connection is not direct. In terms of N, the sources of N that affect rice growth are mainly from soil (the indicator we choose is Alkali-hydrol N) and N application rate, and these N directly affect the growth process and the final yield of rice. Researchers have done a lot of studies in dealing with the relationship between the four aspects. Based on these studies, our approach is to directly perform regression analysis between apparent N supply of field (Nx) (apparent N supply of soil, that is, the content of soil Alkali-hydrol N + N appliation rate) and yield. By this way, we can know how much target yield corresponds to the Nx is, that is apparent total N uptake (Nz). At the same time, in order to avoid the chemical determination of soil Alkali-hydrol N, we estimate the apparent N supply of soil by rice growth status (Ys,SPAD value derivative index). Specifically, the relationship between the Ys and the Nx is established. Through the above two relationships, when the Ys is obtained during the rice growth process, the Nx and apparent N supply of soil can be obtained. The difference between Nz and apparent N supply of soil is the data for manual decision on N.

Reviewer #2: Please add some more parameters in relation to rice yield components and their responses to N and SPAD values.

Response: Referring to the above reply, in this manuscript, the rice yield composition and its response to N and SPAD values are not the focus we want to express.

Reviewer #2: If possible, please add some experimental photos of SPAD values with changing N application rates.

Response: We are sorry. No photos of SPAD values with changing N application rates were not taken during the field experiment.

Reviewer #2: The language needs to be improved further.

Response: The manuscript language has been improved again.

Reviewer #2: The description of results “effect of variable N application based on the leaf value model” is too easy.

Response: It has been supplemented in the manuscript.

Reviewer #2: In Table 3, please explain what are Nw, Ys and Ng.

Response: They have been added explanation in Table 3.

Reviewer #2: In Table 4, please explain how to calculate Yield, Harvest index, N agronomic efficiency, N recovery efficiency, N partial productivity and N contribution.

Response: They have been added explanation in Table 4. At the some time, They are also clearly described in ‘materials and methods’.

Reviewer #2: Conclusions are too easy.

Response: It has been supplemented in the manuscript.

Reviewer #2: Statistical Analysis is too easy, please add some more description.

Response: It has been supplemented in the manuscript.

Reviewer #2: Please separate “Experiment Design” into several parts including such as experimental design, soil and crop management, plant sampling, calculation, to make the materials and methods more clear. 

Response: The ‘materials and methods’ section have been revised.

Thank two reviewers again for their recognition and valuable suggestions.

---

## [Decision Letter · Decision Letter 1]

12 May 2020

Stability and applicability of the leaf value model for variable rate fertilization based on SPAD value in rice

PONE-D-20-02735R1

Dear Dr. feng,

We are pleased to inform you that your manuscript has been judged scientifically suitable for publication and will be formally accepted for publication once it complies with all outstanding technical requirements.

With kind regards,

Bing Xue, Ph.D.

Academic Editor

PLOS ONE

Additional Editor Comments (optional):

Reviewers' comments:

Reviewer's Responses to Questions

**Comments to the Author**

1. If the authors have adequately addressed your comments raised in a previous round of review and you feel that this manuscript is now acceptable for publication, you may indicate that here to bypass the “Comments to the Author” section, enter your conflict of interest statement in the “Confidential to Editor” section, and submit your "Accept" recommendation.

Reviewer #2: All comments have been addressed

2. Is the manuscript technically sound, and do the data support the conclusions?

Reviewer #2: Yes

3. Has the statistical analysis been performed appropriately and rigorously? 

Reviewer #2: Yes

4. Have the authors made all data underlying the findings in their manuscript fully available?

Reviewer #2: Yes

5. Is the manuscript presented in an intelligible fashion and written in standard English?

Reviewer #2: Yes

6. Review Comments to the Author

Reviewer #2: Dear authors, I am satisfied with the revision. Accept. Thank you very much for your revision. Best wishes!

7. PLOS authors have the option to publish the peer review history of their article (what does this mean?). If published, this will include your full peer review and any attached files.

Reviewer #2: No

---

## [Editor Report · Acceptance letter]

22 May 2020

PONE-D-20-02735R1 

Stability and applicability of the leaf value model for variable nitrogen application based on SPAD value in rice

Dear Dr. feng:

I am pleased to inform you that your manuscript has been deemed suitable for publication in PLOS ONE. Congratulations! Your manuscript is now with our production department. 

With kind regards,

on behalf of

Professor Bing Xue 

Academic Editor

PLOS ONE